# A Novel Fault Diagnosis Method for Rolling Bearing Based on Hierarchical Refined Composite Multiscale Fluctuation-Based Dispersion Entropy and PSO-ELM

**DOI:** 10.3390/e24111517

**Published:** 2022-10-24

**Authors:** Yinsheng Chen, Zichen Yuan, Jiahui Chen, Kun Sun

**Affiliations:** 1School of Measurement and Communication Engineering, Harbin University of Science and Technology, Harbin 150080, China; 2National Experimental Teaching Demonstration Center of Measurement and Control Technology and Instrumentation, Harbin University of Science and Technology, Harbin 150080, China

**Keywords:** rolling bearing fault diagnosis, feature extraction, hierarchical refined composite multiscale fluctuation-based dispersion entropy (HRCMFDE), particle swarm optimization-based extreme learning machine (PSO-ELM), load migration

## Abstract

This paper proposes a novel fault diagnosis method for rolling bearing based on hierarchical refined composite multiscale fluctuation-based dispersion entropy (HRCMFDE) and particle swarm optimization-based extreme learning machine (PSO-ELM). First, HRCMFDE is used to extract fault features in the vibration signal at different time scales. By introducing the hierarchical theory algorithm into the vibration signal decomposition process, the problem of missing high-frequency signals in the coarse-grained process is solved. Fluctuation-based dispersion entropy (FDE) has the characteristics of insensitivity to noise interference and high computational efficiency based on the consideration of nonlinear time series fluctuations, which makes the extracted feature vectors more effective in describing the fault information embedded in each frequency band of the vibration signal. Then, PSO is used to optimize the input weights and hidden layer neuron thresholds of the ELM model to improve the fault identification capability of the ELM classifier. Finally, the performance of the proposed rolling bearing fault diagnosis method is verified and analyzed by using the CWRU dataset and MFPT dataset as experimental cases, respectively. The results show that the proposed method has high identification accuracy for the fault diagnosis of rolling bearings with varying loads and has a good load migration effect.

## 1. Introduction

As a highly standardized precision mechanical device, rolling bearings are widely used in aerospace, wind power generation, automotive railways, and industrial production [1]. It is one of the most common components in rotating machinery and mainly serves as a connection and support [2]. However, some studies have shown that 45% to 55% of rotating machinery faults are caused by rolling bearings [3]. The occurrence of these faults is highly likely to cause catastrophic emergencies and severe loss of life and property [4]. Early identification and severity estimation of rolling bearing fault is the focus and difficulty of condition monitoring of rotating machinery. Therefore, it is necessary to conduct in-depth research on the fault generation mechanism and fault diagnosis methods and select automatic, systematic, and intelligent signal processing and feature extraction methods to improve the operational reliability and health management level of rolling bearings. The study of rolling bearing fault diagnosis is conducive to solving the problems of equipment maintenance and economic efficiency in the final industrial application. It is of great significance in ensuring the regular operation of rotating machinery.

The essence of rolling bearing fault diagnosis based on vibration signal features is a pattern identification process, which mainly includes feature extraction and fault classification [5]. As the vibration signal characteristics are closely related to the physical structure of the bearing, the vibration signal often contains rich information about the fault state of the rolling bearing [6,7]. At present, in the rolling bearing condition of monitoring and fault diagnosis technology, vibration signal analysis is one of the current widely used technical means.

As one of the critical technologies for rolling bearing fault diagnosis, the result of fault feature extraction directly affects the final fault identification accuracy. Therefore, selecting a method that can maximize the useful information without loss and have an excellent clustering of the extracted fault features is one of the crucial challenges faced by research in this field in recent years. Additionally, for nonlinear time series, the feature extraction method based on signal complexity is very important. There are two typical time series complexity indicators, one is Lempel-Ziv complexity (LZC), and the other is entropy. LZC [8] is typically used to evaluate the disorder of a given length sequence and is essentially based on coarse-grained processes, the simplest method being conversion to binary sequences (zeros and ones). However, LZC is susceptible to human factors. Therefore, in recent years, many experts and scholars have improved the traditional LZC algorithm [9]. Mao et al. [10] combined dispersion entropy (DE) with LZC to propose a new metric, dispersion Lempel-Ziv complexity (DLZC), and applied it to the biomedical field with good results. Li et al. [11] firstly added fluctuation information to DLZC. They proposed fluctuation-based DLZC (FDLZC), and then introduced an improved coarse-grained operation to propose refined composite multiscale FDLZC (RCMFDLZC), which was applied to both bearing fault diagnosis and ship signal classification with a high recognition rate.

Entropy, as a measure of time series uncertainty and irregularity [12], is widely used to extract the dynamic characteristics of rotating machinery because of its advantages, such as good clustering capability and classification accuracy [13]. Ali et al. [14] used an empirical mode decomposition (EMD) combined with energy entropy for feature vector extraction to automatically monitor bearing fault severity. However, EMD is prone to modal mixing and endpoint effect, which will affect the result of signal decomposition to a certain extent. Li et al. [15] proposed a new method for rolling bearing fault feature extraction, which is composed of two simple methods: ensemble empirical mode decomposition (EEMD) and improved frequency band entropy (IFBE), which can extract the early weak fault features of rolling bearing effectively. It can achieve accurate diagnosis of rolling bearing. Although EEMD solves the modal mixing problem of EMD by adding white Gaussian noise (WGN), the computation time increases as the number of iterations increases. Liu et al. [16] proposed the method of extracting feature vectors using local mean decomposition (LMD) and multiscale entropy (MSE) to identify different bearing operating conditions. Although LMD has improved the proper method to make the envelope function at the endpoints closer to the actual situation, the computational efficiency is lower. Wavelet packet transform (WPT) is also a typical time-frequency analysis method for rolling bearing vibration signals. Chen et al. [17] generated feature vectors describing different fault types of rolling bearings by calculating the energy entropy values of varying frequency band sub-signals obtained from WPT decomposition. However, WPT decomposition needs to determine the number of wavelet basis functions and the number of WPT decomposition layers before use, which affects the adaptive capability of the method. Variational mode decomposition (VMD) is a new multi-component signal decomposition method based on Wiener filtering and Hilbert transform. Ding et al. [18] applied VMD to the feature extraction of gear fault vibration signals by using the sample entropy (SE) values of several intrinsic mode functions (IMFs) decomposed by VMD as fault features. However, the decomposition effect of VMD is also primarily influenced by parameter selection. With the successive introduction of various improved time-frequency analysis methods, the issue of practical efficiency and application scope cannot be ignored. Therefore, the operability and accuracy of feature extraction methods have also become a particular focus and research direction for many scholars, and some of them began trying to optimize the decomposition step and improve it by incorporating it into the process of entropy calculation [19,20].

The above common entropy measures still have some limitations. The approximate entropy (ApEn) proposed by Pincus [21] leads to less stable results due to the excessive dependence on the length of the time series. The SE proposed by Richman et al. [22] is computationally inefficient and unsuitable for analyzing long-time series. Feature extraction based on permutation entropy (PE) [23] ignores the influence of the amplitude of the elements in the time series on the entropy value, making the extracted fault features more random, which in turn, affects the accuracy of subsequent fault identification. To solve the problems of PE, Hamed et al. [24] proposed amplitude-aware permutation entropy (AAPE) and effectively improved the amplitude and frequency sensitivity of PE to time series. Given this, Chen et al. [25] combined AAPE and multiscale entropy algorithm, and proposed multiscale amplitude aware permutation entropy (MAAPE) to extract the fault characteristics of rolling bearings, but when the scale factor in MAAPE is high, the number of elements in the coarse-grained time series decreases, which will seriously affect the stability of the entropy measure. To solve the above problem, Rostaghi and Azami [26] proposed a new, more stable, and computationally efficient entropy measure called dispersion entropy (DE) in 2016. Based on DE, further considering the fluctuation of the signal, they proposed fluctuation-based dispersion entropy (FDE) in 2018 [27]. FDE generates different fluctuation dispersion patterns by mapping each element of the measurement sequence to another class, which not only gives FDE a strong anti-noise capability but also avoids the defect of losing amplitude information in PE.

However, the FDE only describes the complexity of nonlinear time series at a single scale, resulting in the loss of a large amount of important information. Azami et al. [28] further carried out a multiscale expansion of the FDE and proposed the multiscale fluctuation-based dispersion entropy (MFDE). However, traditional coarse-grained methods do not consider the relationship between different coarse-grained time series, and the entropy value will become unstable as the scale factor increases [29]. Gan et al. [30] proposed the composite multiscale fluctuation-based dispersion entropy (CMFDE) and demonstrated that the coarse-grained sliding algorithm could improve the stability of MFDE. Subsequently, Zhou et al. [31] utilized the refined composite multiscale fluctuation-based dispersion entropy (RCMFDE), which is the same as the CMFDE coarse-grained method and has similar principles. Unlike CMFDE, which calculates the mean entropy values of different coarse-grained time series, RCMFDE adopts the refined composite method. However, both MFDE and RCMFDE focus only on the low-frequency features of the time series but ignore the equally crucial high-frequency information [32]. Ke et al. [33] proposed hierarchical fluctuation-based dispersion entropy (HFDE). Although HFDE can accurately describe the low and high frequency components of the signal by constructing its characteristics in different frequency bands to achieve accurate and comprehensive measurements, it has low utilization of the elements in the original time series and is easy to cause information loss. At the same time, the stability of HFDE will be affected by the increase in the number of layers, which is not conducive to the extraction of bearing fault features. Thus, this paper combines the hierarchical theory algorithm with RCMFDE and proposes the hierarchical refined composite multiscale fluctuation-based dispersion entropy (HRCMFDE). This feature extraction method combines the advantages of HFDE and RCMFDE, avoids their disadvantages, and not only solves the problem of missing high-frequency signals but also has better stability. Applying it to the feature extraction part of rolling bearing fault diagnosis can describe the fault features more accurately and help to improve the accuracy of subsequent fault identification.

Fault type identification is another important part of rolling bearing fault diagnosis. It is crucial to choose a suitable high-performance classifier to ensure identification accuracy. Commonly used methods for fault type identification include k-nearest neighbor (k-NN) classification algorithm, support vector machines (SVM), artificial neural network (ANN), and recurrent neural network (RNN), etc. The KNN algorithm performs pattern recognition on the testing samples by referring to the classes of the nearest sample. Tian et al. [34] used spectral kurtosis to extract fault features and combined principal component analysis (PCA) and k-NN classification algorithm for fault diagnosis of motor bearings. Because the k-NN classification algorithm requires distance calculation for all samples at each prediction, it has the disadvantage of large computational effort and slow classification speed. Zheng et al. [35] used generalized refined composite multiscale fuzzy entropy (GRCMFE) for feature extraction and multi-cluster feature selection for supervised learning, and the gravitational search algorithm optimized support vector machine (GSA-SVM) for fault pattern recognition. Still, the fault diagnosis results are greatly influenced by the choice of kernel parameters in the classifier. Wen et al. [36] proposed a hierarchical convolutional neural network (HCNN) structure containing two classifiers, and this two-level hierarchical diagnostic network can simultaneously estimate the fault mode and severity to achieve the diagnosis of bearing faults. Still, the method has bottlenecks in terms of slow gradient learning speed and preset parameters.

Extreme learning machine (ELM) [37], as a novel classification and identification method, has the advantages of fast learning speed and good generalization performance compared with traditional classification methods. Ye et al. [38] introduced singular value vectors as fault feature vectors into ELM for the identification and classification of bearing faults, to reduce manual intervention and shorten fault diagnosis time. Mao et al. [39] proposed an online timing prediction diagnosis method for unbalanced faults based on ELM for the problem that the actual bearing fault data are unbalanced and much less in number than normal data. However, there are uncertainties in the input weights and hidden layer neuron thresholds predetermined in the ELM classifier, which require further optimization to improve the classification accuracy. Particle swarm optimization (PSO) [40] has been widely used as an effective technique to search for global minima. The PSO does not have complex evolutionary operators and requires fewer parameters to be tuned [41]. Therefore, the mixture of PSO and linear mapping has a promising application in the training of feedforward neural networks [42]. In the field of rolling bearing fault diagnosis, the PSO algorithm has been applied in both feature extraction and fault identification. He et al. [43] used the PSO algorithm to optimize the number of components and the penalty factor in VMD, so that the decomposed IMF contains richer fault information. Chen et al. [44] used the PSO algorithm to find the most suitable input weights and hidden layer neuron thresholds for ELM to obtain more accurate fault identification results. In this paper, the PSO algorithm is utilized in the latter application.

In this paper, a rolling bearing fault diagnosis method combining HRCMFDE and PSO-ELM is proposed, and its main contributions are summarized as follows:A novel fault feature extraction method based on HRCMFDE is proposed. The method quantifies the high-frequency and low-frequency features of the measured time series by introducing the hierarchical theory algorithm, which effectively overcomes the problem of high-frequency information loss caused by the coarse-grained process. Meanwhile, HRCMFDE maps each element of the time series to different classes and generates different fluctuation dispersion patterns, which not only has strong anti-noise capability but also avoids the defect of losing amplitude information.The performance of the proposed rolling bearing fault diagnosis method is verified using two typical rotating machinery fault datasets. The experimental results show that the proposed fault diagnosis method can not only accurately identify the fault types with varying loads, but also have a high fault identification effect even under load migration.

The rest of this paper is organized as follows. In Section 2, the basic principles of the proposed HRCMFDE rolling bearing fault feature extraction method and PSO optimized ELM fault identification method are briefly introduced. Section 3 details the algorithm flow of the proposed HRCMFDE and PSO-ELM rolling bearing fault diagnosis method. The proposed method is experimentally verified by using two typical rotating machinery fault datasets, and the experimental results are analyzed and discussed in Section 4. Finally, Section 5 summarizes the research content and results of this paper and prospects the future research content.

## 2. Methodologies

### 2.1. Feature Extraction

#### 2.1.1. Multiscale Fluctuation-Based Dispersion Entropy (MFDE)

FDE is a method for calculating the complexity of nonlinear time series based on Shannon’s entropy and fluctuation dispersion patterns. In the calculation, FDE considers the difference of discrete spectra of adjacent elements, and can easily distinguish deterministic signals from random signals by comparing fluctuation-based dispersion patterns, so it is widely used for feature extraction of nonlinear and non-stationary signals. However, FDE can only quantify the dynamic features of the measured series on a time scale, therefore, it cannot capture long correlations. To solve this problem, Azami et al. proposed MFDE, a multiscale entropy (MSE) improvement method for FDE. MFDE has higher computational efficiency and better fault feature description capability than MSE and MFE. The basic principle of MFDE is as follows:

For a given univariate signal $u=\left\{u_{1},u_{2},\cdots ,u_{L}\right\}$ of length L, the signal u is divided into a set of non-overlapping segments of length τ, where τ is the scale factor. Afterward, the average of each non-overlapping segment is calculated to derive a coarse-grained time series as follows:(1)$$u_{j}^{\tau }=\frac{1}{\tau }\sum_{b=(j-1)\tau +1}^{j\tau }u_{b},1\leq j\leq \left[L/\tau \right]=N$$

Finally, the FDE of each coarse-grained signal $u_{j}^{\tau }$ is calculated. The FDE calculation procedure for time series $u_{j}^{\tau }$ is as follows:

Mapping the coarse-grained time series $u_{j}^{\tau }=\left\{x_{1},x_{2},\cdots ,x_{N}\right\}$ of length N to the time series $y=\left\{y_{1},y_{2},\cdots ,y_{N}\right\}$ through the normal cumulative distribution function (NCDF) in Equation (2):(2)$$y_{j}=\frac{1}{\sigma \sqrt{2\pi }}{\int_{-\infty }^{x_{j}}e}^{\frac{-\left(t-\mu \right)}{2\sigma^{2}}}dt$$
where σ and μ are the standard deviation (SD) and mean of the time series, respectively. Then, each $y_{j}$ is linearly assigned to an integer from 1 to c. For that reason, for each member of the mapped signal y, Equation (3) is used to map $y_{j}$ to $z_{j}^{c}$:(3)$$z_{j}^{c}=round\left(c\cdot y_{j}+0.5\right)$$
where $z_{j}^{c}$ denotes the jth member of the classified time series $z_{i}^{m,c}$, c is an integer and round(·) is a rounding function. Time series $z_{i}^{m,c}$ are defined concerning embedding dimension m-1 and time delay d according to $z_{i}^{m,c}=\left\{z_{i}^{c},z_{i+d}^{c},\cdots ,z_{i+(m-1)d}^{c}\right\}$, $i=1,2,\cdots ,N-\left(m-1\right)d$.

Each time series $z_{i}^{m,c}$ is mapped to a fluctuation-based dispersion pattern $\pi_{\nu_{0}\nu_{1}\cdots \nu_{\left(m-1\right)}}$, where $z_{i}^{c}=v_{0},z_{i+d}^{c}=v_{1},\cdots ,z_{i+(m-1)d}^{c}=v_{m-1}$. The number of possible fluctuation-based dispersion patterns that can be assigned to each time series $z_{i}^{m,c}$ is equal to ${\left(2c-1\right)}^{m-1}$. For each ${\left(2c-1\right)}^{m-1}$ potential dispersion pattern $\pi_{\nu_{0}\nu_{1}\cdots \nu_{\left(m-1\right)}}$, relative frequency is obtained as follows:(4)$$p\left(\pi_{\nu_{0}\nu_{1}\cdots \nu_{\left(m-1\right)}}\right)=\frac{num\left\{i\left|i\leq N-\left(m-1\right)d,z_{i}^{m,c}has\;type\pi_{\nu_{0}\nu_{1}\cdots \nu_{\left(m-1\right)}}\right.\right\}}{N-(m-1)d}$$
where num(·) means cardinality. In fact, $p\left(\pi_{\nu_{0}\nu_{1}\cdots \nu_{\left(m-1\right)}}\right)$ shows the number of dispersion patterns of $\pi_{\nu_{0}\nu_{1}\cdots \nu_{\left(m-1\right)}}$ that is assigned to $z_{i}^{m,c}$, divided by the total number of embedded signals with embedding dimension m.

Finally, based on Shannon’s definition of entropy, the FDE value is calculated as follows:(5)$$FDE(x,m,d,c)=-\sum_{\pi =1}^{{\left(2c-1\right)}^{m-1}}p\left(\pi_{\nu_{0}\nu_{1}\cdots \nu_{\left(m-1\right)}}\right)\cdot ln\;p\left(\pi_{\nu_{0}\nu_{1}\cdots \nu_{\left(m-1\right)}}\right)$$

#### 2.1.2. Refined Composite Multiscale Fluctuation-Based Dispersion Entropy (RCMFDE)

The multiscale algorithm obtains a new coarse-grained sequence by cutting the original sequence equally spaced and calculating the average of short sequences. However, if the starting point is different, the entropy value will also fluctuate to a certain extent, and considering that the relationship between the new coarse-grained sequence elements cannot be neglected, further refinement of MFDE by a refined composite algorithm is required. RCMFDE replaces the traditional coarse-grained method by sliding coarse-grained data processing, effectively avoiding the instability of entropy values caused by increasing scale factors, and shortening coarse-grained time series.

The probability mean of the fluctuation dispersion pattern $\pi_{\nu_{0}\nu_{1}\cdots \nu_{\left(m-1\right)}}$ in the coarse-grained series $y_{j}^{s}(1\leq s\leq \tau )$ of the original time series $x=\left\{x_{1},x_{2},\cdots ,x_{N}\right\}$ can be calculated as:(6)$$\bar{p}\left(\pi_{\nu_{0}\nu_{1}\cdots \nu_{\left(m-1\right)}}\right)=\frac{1}{\tau }\sum_{1}^{\tau }p_{s}^{\tau }$$
where $p_{s}^{\tau }$ is the frequency of the dispersion mode $\pi_{\nu_{0}\nu_{1}\cdots \nu_{\left(m-1\right)}}$ in the coarse-grained sequence $y_{j}^{s}$.

For the maximum scale factor $\tau_{max}$, the RCMFDE value is defined as the Shannon entropy of the average dispersion model obtained after the time series is advected:(7)$$RCMFDE(x,m,d,c,\tau_{max})=-\sum_{\pi =1}^{{\left(2c-1\right)}^{m-1}}\bar{p}\left(\pi_{\nu_{0}\nu_{1}\cdots \nu_{\left(m-1\right)}}\right)\cdot ln\;\bar{p}\left(\pi_{\nu_{0}\nu_{1}\cdots \nu_{\left(m-1\right)}}\right)$$

#### 2.1.3. Hierarchical Refined Composite Multiscale Fluctuation-Based Dispersion Entropy (HRCMFDE)

Basic Principle

The multiscale analysis only captures the low-frequency information of the measured sequence. In contrast, part of the high-frequency information is discarded, and there is a loss of amplitude information, which will have a certain impact on entropy. Therefore, this paper introduces the hierarchical theory into RCMFDE, and the principle of hierarchical decomposition is shown in Figure 1. By constructing high-frequency and low-frequency operators, the high-frequency and low-frequency features of the measured sequence are quantified simultaneously. The HRCMFDE proposed in this paper evaluates the dynamic features hidden in the low-frequency and high-frequency components of the vibration signal, thus overcoming the problem of amplitude information loss due to the coarse-grained process.

Define two symbols $Q_{0}$ and $Q_{1}$, which represent the averaging operator and the difference operator, respectively. For time series $x=\left\{x_{1},x_{2},\cdots ,x_{N}\right\}$ of length $2^{n}$, the low-frequency features averaging operators $Q_{0}\left(x\right)$ and the high-frequency operators $Q_{1}\left(x\right)$ are:(8)$$Q_{0}\left(x\right)=\frac{x_{2j}+x_{2j+1}}{2},j=1,2,\cdots ,2^{n-1}$$
(9)$$Q_{1}\left(x\right)=\frac{x_{2j}-x_{2j+1}}{2},j=1,2,\cdots ,2^{n-1}$$

The operator $Q_{j}$ is in matrix form and can be expressed as:(10)$$Q_{j}={\left[\begin{array}{ccccccc} \frac{1}{2} & \frac{{\left(-1\right)}^{j}}{2} & 0 & 0 & \cdots & 0 & 0\\ 0 & 0 & \frac{1}{2} & \frac{{\left(-1\right)}^{j}}{2} & \cdots & \vdots & \vdots \\ \vdots & \vdots & \vdots & \vdots & \cdots & 0 & 0\\ 0 & 0 & 0 & 0 & \cdots & \frac{1}{2} & \frac{{\left(-1\right)}^{j}}{2} \end{array}\right]}_{2^{n-1}\times 2^{n}}$$
where $j\in \left[0,1\right]$. When the layer number is n, construct vector $\gamma =[\gamma_{1},\gamma_{2},\cdots ,\gamma_{n}]\in \{0,1\}$. The hierarchical node number e can be calculated by the following equation:(11)$$e=\sum_{j=1}^{k}\gamma_{j}2^{k-j}$$

Based on the vector γ, Equations (8) and (9) are repeated to obtain the hierarchical component $x_{n,e}$ corresponding to the eth node of the nth layer.
(12)$$x_{n,e}=Q_{\gamma_{n}}^{n}\cdot Q_{\gamma_{n-1}}^{n-1}\cdot \cdots Q_{\gamma_{1}}^{1}\cdot x$$

Then, the HRCMFDE of the original signal x can be obtained by the following equation:(13)$$HRCMFDE(x,m,d,c,\tau_{max},n)=Average(RCMFDE(x,m,d,c,\tau_{max}))$$
where Average(·) denotes the calculation of the arithmetic mean.

The calculation flowchart of HRCMFDE is shown in Figure 2.

Parameter Selection

The performance of HRCMFDE is determined by five parameters: embedding dimension m, the number of classes c, time delay d, maximum scale factor $\tau_{max}$, and number of hierarchical layers n. The choice of parameters directly affects the effectiveness of HRCMFDE in measuring time series.

1.Embedding dimension m.

If m is too small, HRCMFDE cannot accurately observe the dynamic changes of the nonlinear time series. Conversely, if m is too large, HRCMFDE cannot detect small changes. According to the references [26], the value range of m is 2–5.

2.The number of classes c.

If c is too small, two different amplitudes may be assigned to the same class and if c is too large, HRCMFDE is sensitive to noise. According to the references [27], the value range of c is an integer in 3–9.

3.Time delay d.

The effect of d on HRCMFDE is small, and some frequency information may be lost when d>1, which is usually taken as the smallest positive integer 1.

4.Maximum scale factor $\tau_{max}$.

If $\tau_{max}$ is too small, HRCMFDE cannot fully extract the features of the nonlinear time series. If $\tau_{max}$ is too large, it is easy to produce unstable and unreliable entropy values. In addition, a larger $\tau_{max}$ may reduce the computational efficiency of HRCMFDE. Therefore, to obtain reliable results, according to reference [28], the maximum scale factor $\tau_{max}$ is usually chosen to be 20, which is sufficient to analyze the time series efficiently.

5.Number of hierarchical layers n.

For the selection of n, the larger n is, the shorter the time series of the corresponding layers, which will lead to insufficient extraction of fault information. However, if n is too small, the decomposition of the original time series is incomplete, and the dimensionality of the extracted features is low and insufficient. According to the references [45,46], n is generally set to 3.

Simulation Signal Analysis

In this section, to discuss the sensitivity of HRCMFDE to the embedding dimension m and the number of classes c, WGN noise, and 1/f noise were used as simulation signals for parametric analysis. Each simulation signal is a randomly generated set of 20 with mean 0, variance 0, and data length 3000, respectively. The time domain waveform and frequency spectrum of a set of random signals are shown in Figure 3. As can be seen from the figure, 1/f noise is less smooth than WGN, which has higher uncertainty than 1/f noise.

To explore the effect of embedding dimension m and the number of classes c on HRCMFDE, two types of noise with different values were discussed. The mean and SD of each node were calculated with m and c as univariate variables, respectively. From the previous section, the range values of m is 2–4, and the three cases of m=2, m=3, m=4 are chosen; the range values of c is 3–9, and the three cases of c=3, c=6, c=9 are chosen.

The effect of the embedding dimension m is shown in Figure 4. Both in WGN and 1/f noise, the mean curve at m=2 is smoother, and the SD value is the smallest. Considering the effect of the degree of dispersion of the 20 sets of noisy data, the CVs (coefficients of variation) were discussed for m at the three cases. By calculating the ratio of SD to mean, the magnitude of CVs is inversely proportional to the computational stability, as shown in Table 1 and Table 2, which more objectively indicates the superiority of m=2 in HRCMFDE.

As shown in Figure 5, it can be visualized that both for WGN and 1/f noise, the mean fluctuation is the smallest when c=6, whereas the SD curve is the most cluttered when c=3, and the SD curves are closer when c=6 and c=9. Further, the calculated CVs are shown in Table 3 and Table 4, respectively. When c is taken as 6 and 9, it is not apparent which value is superior, and it can be seen that its influence on the stability of entropy is not significant. To ensure consistency, c=6 is taken uniformly in the subsequent experiments of this paper.

To verify the superiority of the proposed HRCMFDE algorithm over the pre-improvement algorithm in terms of stability, four different derivative algorithms for the fluctuation dispersion entropy were also compared in this paper, and the parameter selection in the experiment is shown in Table 5.

Due to the different principles of each algorithm, the extracted feature dimensions also differ, so the experiments are divided into two groups for comparison, one group is HFDE and HRCMFDE, as shown in subplot (a) and subplot (b) in Figure 6, and the feature dimension of the horizontal coordinate is the number of hierarchical nodes; the other group is MFDE and RCMFDE, as shown in subplot (c) and subplot (d) in Figure 6, and the feature dimension of the horizontal coordinate is the scale factor size.

As shown in Figure 6, on the whole, it is not difficult to find that the fold trends for the four different entropies, both at WGN and 1/f noise signal, are roughly similar, which shows that the type of noise signal has little effect on the stability of the entropy value calculation. Observing subplot (c) and subplot (d), where the folds both show a decreasing trend, this is because the single coarse granulation process in MFDE and RCMFDE leads to incomplete information, which makes the entropy of each scale significantly different. Then, by comparing the volatility of the folds in subplot (a) and subplot (b), it can be found that the stability of HFDE is not as good as that of HRCMFDE in both noises. This is due to the introduction of sliding coarse-grained, which makes the information considered by HRCMFDE more complete and computationally more stable compared to HFDE. In addition, comparing subplots (a), (c), and subplots (b), (d), respectively, it is found that the skewness of RCMFDE is significantly enhanced compared with that of MFDE, reflecting that the stability of entropy is improved after the refined composite algorithm improvement of MFDE. However, there are still some slight deficiencies compared with the HRCMFDE proposed in this paper. This affirms, to some extent, the introduction of hierarchical theory in solving the problem of information loss caused by the coarse-grained process. In summary, it can be concluded that HRCMFDE has the lowest error rate in measuring nonlinear dynamic changes and has more considerable superiority compared with MFDE, HFDE, and RCMFDE.

### 2.2. Fault Identification

#### 2.2.1. Extreme Learning Machine (ELM)

The ELM algorithm is a single-hidden layer feedforward neural network (SLFN) learning algorithm proposed by Huang et al. Its distinctive feature is that the input weights and biases of the hidden layer nodes are randomly generated using a random algorithm, and the only variable that needs to be computed is the output weight obtained by the least squares. As a result, it can learn faster than traditional intelligent algorithms and perform better generalization [47].

Generally, an ELM network includes three parts which are an input layer, a hidden layer, and an output layer. Let m, l, and n be the number of nodes in the input, implicit, and output layers of the network, respectively, and g(x) be the activation function of the neurons in the implicit layer. Set there are N different samples $\left(x_{i},t_{i}\right)$, 1≤i≤N, where:(14)$$x_{i}={\left[x_{i1},x_{i2},\cdots ,x_{im}\right]}^{T}\in R^{m}$$
(15)$$t_{i}={\left[t_{i1},t_{i2},\cdots ,t_{in}\right]}^{T}\in R^{n}$$

The network structure of ELM is shown in Figure 7. The network model of ELM can be expressed mathematically as follows:(16)$$\sum_{j=1}^{l}\beta_{j}g\left(\omega_{j}\cdot x_{i}+b_{j}\right)={ο}_{i}$$
where j=1,⋯,N. $\omega_{j}={\left[\omega_{j1},\omega_{j2},\cdots ,\omega_{jn}\right]}^{T}$ is the input weight that connects the jth hidden node with the input nodes, $\beta_{j}={\left[\beta_{j1},\beta_{j2},\cdots ,\beta_{jm}\right]}^{T}$ is the output weight that connects the jth hidden node with the output nodes, $b_{j}$ is the bias of the jth hidden node, and $\omega_{j}\cdot x_{i}$ is the inner product of $\omega_{j}$ and $x_{i}$.

When the above standard SLFN infinitely approximates the N samples and the error is equal to 0, i.e., $\sum_{i=1}^{N}\left\|{ο}_{i}-t_{i}\right\|=0$, there exist $\beta_{j}$, $\omega_{j}$, $b_{j}$, so that:(17)$$\sum_{j=1}^{l}\beta_{j}g\left(\omega_{j}\cdot x_{i}+b_{j}\right)=t_{i}$$

The simplified form of Equation (17) is:(18)Hβ=T
where:(19)$$H\left(\omega_{1},\cdots ,\omega_{l},b_{1},\cdots ,b_{l},x_{1},\cdots ,x_{N}\right)={\left[\begin{array}{ccc} g\left(\omega_{1}\cdot x_{1}+b_{1}\right) & \cdots & g\left(\omega_{l}\cdot x_{1}+b_{l}\right)\\ \vdots & \vdots & \vdots \\ g\left(\omega_{1}\cdot x_{N}+b_{1}\right) & \cdots & g\left(\omega_{l}\cdot x_{N}+b_{l}\right) \end{array}\right]}_{N\times l}$$
(20)$$\beta ={\left[\begin{array}{c} \beta_{1}^{T}\\ \vdots \\ \beta_{l}^{T} \end{array}\right]}_{l\times m}$$
(21)$$T={\left[\begin{array}{c} t_{1}^{T}\\ \vdots \\ t_{N}^{T} \end{array}\right]}_{N\times m}$$
where H is the output matrix of the hidden layer in ELM.

The difference between the ELM algorithm and the general SLFN algorithm is that the input weight ω and hidden layer bias b of the former are given randomly at the beginning of the algorithm and accordingly calculate the output matrix H. Then, what needs to be done is to determine the parameter β. Training the feedforward neural network can be regarded as seeking out the solution of Hβ=T by the least-square method. The output weight matrix β can be obtained by solving this equation:(22)$$\beta =H^{+}T$$
where $H^{+}$ is the Moore–Penrose generalized inverse of H.

#### 2.2.2. Particle Swarm Optimization-Based Extreme Learning Machine (PSO-ELM)

Although ELM has the advantages of easy parameter selection, fast learning speed, good generalization performance, and does not fall into local optimum, its classification performance will be affected because the input weights and hidden layer neuron thresholds are randomly generated during training. Therefore, in this paper, the PSO algorithm is used to optimize the network structure by seeking the input weights and hidden layer thresholds of ELM, so that the classification accuracy of ELM can be higher. The flow chart of the PSO-ELM algorithm is shown in Figure 8 below.

## 3. Proposed Method

In this paper, a novel rolling bearing fault diagnosis method based on the combination of HRCMFDE and PSO-ELM is proposed, and the flowchart of the proposed method is shown in Figure 9. The main processes of the rolling bearing fault diagnosis method are described below.

### 3.1. Data Preprocessing

Firstly, the data acquisition system is used to obtain sufficient raw vibration signals of rolling bearings in different fault states. Then, each original vibration signal is segmented by length N to obtain a set of equal-length samples for that fault state. Finally, a sample set is constructed using samples from different fault states, and this sample set is divided into a training sample set and a test sample set.

### 3.2. Training Process

Firstly, feature extraction is performed for each sample in the training set. HRCMFDE performs n-layer hierarchical decomposition of the sample signal to obtain $2^{n}$ hierarchical decomposition signals. The RCMFDE value is obtained for each hierarchical decomposition signal, and then the signal features of that frequency band are extracted by calculating the arithmetic mean of RCMFDE. The HRCMFDE values of all hierarchical decomposition signals are calculated to form a feature vector, and the feature vectors of all samples in the training set are formed into a training sample feature set. At the same time, to ensure that the fault features extracted by HRCMFDE have better noise suppression capability, the parameters of HRCMFDE are selected using simulated interference signals. Then, the ELM classifier is trained by using the training sample feature set. The input weights and hidden layer neuron thresholds in the ELM classifier are optimized by PSO to obtain optimal fault identification accuracy. Finally, rolling bearing fault diagnosis can be achieved with the obtained PSO-ELM classifier.

### 3.3. Testing Process

Firstly, the HRCMFDE is used to extract the features of the test signal and obtain the feature vector of the test signal. Then, the feature vectors are input to the PSO-ELM classifier. Finally, the results of the rolling bearing fault diagnosis are output.

## 4. Experiments

### 4.1. Case 1: CWRU Dataset

#### 4.1.1. Experiment Setup

In this case, the vibration signals collected at the driving end of the Case Western Reserve University (CWRU) [48] bearing database were selected as experimental data for the experiment. The experimental platform mainly consists of a load motor, accelerometer, torque sensor, and dynamometer. The experimental data set includes vibration signals collected in four fault types: normal (N), ball fault (B), outer race fault (O), and inner race fault (I). The signals collected in each type are classified according to the fault severity and the size of the load. The fault severity is 0.007, 0.014, 0.021, and 0.028 inches, respectively, and the load size range from 0 hp, 1 hp, 2 hp, and 3 hp (1 hp = 746 W).

In this paper, samples of various bearing states with a sampling frequency of 12 KHZ at the driving end were selected for experiments of rolling bearing fault diagnosis, with a total of 12 rolling bearing health states without differentiating the load. The samples included in the CWRU experimental dataset are shown in Table 6, where “√” means the sample exists and is selected, and “*” means the sample does not exist.

Set the data length of each subsample N=2400, take the first 120,000 points of the bearing vibration signals collected in each state (sampling time is 10 s), and divide the bearing vibration signals into 50 groups of 2400 points without overlapping (sampling time is 0.2 s). Each group is a sub-sample, and each state is obtained as a 50 × 2400 data matrix, with each data matrix containing 50 sub-samples.

The preprocessed dataset was canonically named “N” for normal status and “B0070” for ball fault (B) with a fault severity of 0.007 inch. Each load condition experiment contains a total of 600 samples. The training set and testing set were divided according to the ratio of 2:3, and 240 training samples and 360 testing samples were obtained. The four bearing fault types were further divided into 12 health states according to the different fault severity (where the outer race fault is missing 0.028-inch severity samples). The specific division is shown in Table 7 below.

#### 4.1.2. Feature Extraction

As shown in Figure 10, the time domain and spectral diagram of the original signals for the three fault types of the ball fault, outer race fault, and inner race fault with a fault severity of 0.007 inch, respectively. It can be observed from the figure that the original signals are nonlinear, non-stationary signals, in which there are a lot of impulse signals and noise interference, and the fault information exists in each frequency band, so it is difficult to extract the fault features of rolling bearings by observation method.

#### 4.1.3. Fault Identification

In the fault identification section, a total of 12 health states were distinguished for different fault types and different fault severities at the same time. Experimentally, the feature vectors with varying loads were extracted and composed into feature sample sets according to the above feature extraction method. Then, the training set and the testing set were randomly divided in the ratio of 2:3. The number of samples in the training set is 240, containing 20 of each state; the number of samples in the testing set is 360, containing 30 of each state. Finally, the PSO-ELM classifier was used for classification. The above process was repeated ten times for independent experiments, and the average identification accuracy was used as the evaluation index for the performance of the proposed fault diagnosis algorithm. The identification results with load 2 hp are shown in Figure 11, and the average identification accuracy can reach 100%. Therefore, the rolling bearing fault diagnosis method proposed in this paper has a satisfactory fault identification result.

#### 4.1.4. Performance Comparison

To illustrate the effectiveness of the proposed HRCMFDE fault feature extraction method, the features extracted by the HFDE, MFDE, RCMFDE, and HRCMFDE algorithms were identified using the PSO-ELM classifier, respectively, as a way to compare the fault identification results of different feature extraction methods. A set of the training set and testing set was arbitrarily selected with load 0 hp. The fault identification results of different feature extraction methods and PSO-ELM are shown in Figure 12. It can be seen that the accuracy of the HRCMFDE algorithm is the highest among the four feature extraction methods, which can reach 100%. At the same time, the features extracted using the RCMFDE algorithm show a significant improvement in fault identification accuracy over the MFDE algorithm.

Further, a set of the training set and testing set was arbitrarily selected with load 0 hp. The features extracted by the HRCMFDE algorithm were classified using ELM, kernel extreme learning machine (KELM) [49], genetic algorithm optimized extreme learning machine (GA-ELM) [50], and PSO-ELM classifier to compare the effects of different classifiers on the fault identification results when the feature vectors are the same. It is obvious from the figure that determining the kernel function and incorporating the parameter optimization algorithm both help to improve the identification accuracy of the ELM classifier, whereas using the GA and PSO algorithms to optimize the parameters in the ELM classifier can lead to more accurate fault identification results. Among them, the PSO-ELM fault identification method used in this paper has the highest accuracy, and all testing samples can be guaranteed to be correctly classified with load 0 hp.

To verify the generalization of the proposed method in this paper, the feature vectors extracted with varying loads were classified using the PSO-ELM classifier, and each group of experiments was repeated ten times. The average identification accuracy was taken as the fault diagnosis results and recorded. The fault classification results are shown in Table 8, from which it can be seen that the rolling bearing fault diagnosis method based on HRCMFDE and PSO-ELM achieves a relatively high identification accuracy, and its effectiveness and generalization are verified through several experiments.

The experiments also utilize the HFDE, MFDE, RCMFDE, and HRCMFDE algorithms to extract feature vectors with varying loads and input them into the PSO-ELM classifier for fault identification, respectively, to further compare the final result of different feature extraction algorithms for bearing fault diagnosis. The samples were selected with four varying loads, and the experiment was repeated ten times to calculate the average value of the fault identification accuracy. The identification accuracies of different feature extraction methods with four varying loads are shown in Table 9. It can be concluded that the feature vectors extracted using the HRCMFDE algorithm, followed by the PSO-ELM classifier for fault identification, have the desired fault identification accuracy with varying loads.

To further illustrate the performance of the proposed rolling bearing fault diagnosis method, the experiments also compared the average identification accuracy and fault identification time of different classification algorithms with four varying loads, and the results are shown in Table 10. From the experimental results, it can be seen that the KELM algorithm has the shortest identification time, but the lowest identification accuracy. The optimization prolongs the identification time of both GA-ELM and PSO-ELM classifiers, but the identification accuracy is somewhat improved. Among them, the identification time of the PSO-ELM algorithm has a significant advantage over GA-ELM, and the average fault identification accuracy is the highest, reaching 99.91%. The standard deviation is also the smallest, which is only 0.11%.

To verify the superiority of the proposed fault diagnosis method, this paper also compared the identification accuracy of the methods in other references, and the experimental results are shown in Table 11. It can be seen that also using the CWRU dataset for experiments, the method in this paper also has significant diagnostic results when divided into 12 fault states. To ensure the objectivity of the experiment, since the dataset used in reference [11] is different from this paper, the experiment attempted to partially reproduce its method, omitting the feature selection part of it, and conducted several experiments according to the parameters set by the authors to obtain the corresponding average accuracy of fault identification.

#### 4.1.5. Load Migration

To avoid algorithmic limitations caused by selecting the training set and testing set with the same load, the experiments combined varying loads to form the training sets and testing sets to achieve the effect of load migration. As Figure 13 shows the fault identification accuracy of one load as the training set and another different load as the testing set. With load 1 hp as the training set and load 2 hp as the testing set, the fault identification accuracy is up to 100%. With load 0 hp as the training set and load 3 hp as the testing set, the fault identification accuracy is the worst, but it can reach 70.67%.

As Figure 14 shows the fault identification accuracy of two loads as the training set and another different load as the testing set. With load 1 hp and load 3 hp as the training set and load 2 hp as the testing set, the fault identification accuracy is up to 100%. With load 0 hp and load 1 hp as the training set and load 3 hp as the testing set, the fault identification accuracy is the worst, but it can reach 77.50%. Thus, it can be seen that the increase in the number of training set samples can effectively improve fault identification in the testing set.

As Figure 15 shows the fault identification accuracy of two loads as the training set and another two varying loads as the testing set. From the figure, it can be seen that the fault identification accuracy can reach more than 80% for both the training set and testing set matching case. Among them, the fault identification accuracy is up to 100% with load 0 hp and load 1 hp as the training set and load 2 hp and load 3 hp as the testing set. The worst fault identification accuracy is achieved in both cases with load 1 hp, load 2 hp as training sets and load 0 hp, load 3 hp as testing sets and with load 1 hp, load 3 hp as training sets and load 0 hp, load 2 hp as testing sets, but it can also reach 83.33%. Further, it can be concluded that with the increase in the number of samples in the training sets and testing sets, the rolling bearing fault identification accuracy also has significant improvement. It can also be shown that the method proposed in this paper has good generalizability.

### 4.2. Case 2: MFPT Fault Dataset

#### 4.2.1. Experiment Setup

The MFPT (Machinery Fault Prevention Technology) [58] bearing fault dataset provided by the American Society for Machinery Fault Prevention Technology is another typical dataset for rolling bearing fault diagnosis and is one of the essential references to verify the performance of the proposed method. The MFPT dataset consists of three sets of experimental bearing vibration data and three actual fault data. Among them, the three sets of experimental bearing vibration data include baseline bearing data, outer race fault data with various loads, and inner race fault data with various loads.

The baseline dataset contains three files, and the data in each file are obtained by sampling at a frequency of 97,656 sps (sample per second) for 6 s with a load of 270 pounds. There are two types of outer race fault data sets, and one contains three files with the same load, sampling rate, and sampling time as the baseline dataset; the other contains seven files, which are, respectively obtained by sampling at 48,828 sps for 3 s with seven load conditions, including 25, 50, 100, 150, 200, 250, and 300 pounds. The inner race fault dataset contains seven files, respectively, obtained by sampling at 48,828 sps for 3 s with seven load conditions, including 0, 50, 100, 150, 200, 250, and 300 pounds.

The samples included in the MFPT experimental dataset are shown in Table 12. In this paper, all the vibration data of the bearing test stand are selected and divided into three classes according to the fault states and locations. The Class1 is normal (N), which contains three baseline data; the Class2 is outer race fault (O), which contains three outer race data by sampling at 97,656 sps and seven outer race data by sampling at 48,828 sps; and the Class3 is Inner race fault (I), which contains seven inner race data by sampling at 48,828 sps.

To match the other fault data, the baseline data and outer race fault data by sampling at 97,656 sps need to be resampled to 48,828 sps, thus changing the number of data points per file from 585,936 to 292,968. Therefore, the experimental data set consists of the following data points: normal has 878,904 data points, outer race fault has 1,904,292 data points, and inner race fault has 1,025,388 data points.

The data length of each subsample is set to N=2400. The first 864,000 sample points in the normal, the first 1,896,000 sample points in the outer race fault state, and the first 1,020,000 sample points in the inner race fault state were taken, respectively. The bearing vibration signals of different states were neatly partitioned into non-overlapping data of length 2400. Each set was a sub-sample to obtain 360 groups of sub-samples in the normal, 790 groups of sub-samples in the outer race fault state, and 425 groups of sub-samples in the inner race fault state, and they constitute the data matrix of each state, respectively.

The bearing data in each state were randomly divided into the training set and testing set in the ratio of 2:3, and the three classes were labeled with three tags (1,2,3) to represent the different states of the bearing. The specific division is shown in Table 13 below.

#### 4.2.2. Feature Extraction

A set of raw signals in three bearing types (normal, inner race fault, and outer race fault) were selected arbitrarily. As shown in Figure 16, the time domain and the corresponding spectrum diagram are shown.

#### 4.2.3. Fault Identification

In the fault identification section, the feature vectors of each bearing state extracted in the previous section were formed into a feature sample set. Then, the training set and the testing set were randomly divided in the ratio of 2:3. The PSO-optimized ELM classifier was used for bearing states classification and was repeated ten times to obtain the average identification accuracy. As shown below in Figure 17, the identification results are listed, and the identification accuracy can reach 100%. It is enough to prove that the proposed method for classification can achieve the expected results.

#### 4.2.4. Performance Comparison

As shown in Figure 18, an arbitrary set of the training set and testing set were selected, and the PSO-ELM classifier was used to identify the features extracted by the HFDE, MFDE, RCMFDE, and HRCMFDE algorithms, respectively, as a way to compare the fault identification results of different feature extraction algorithms. As seen from the figure, the accuracy of the HRCMFDE algorithm is the highest among the four feature extraction methods, which can also reach 100%.

Further, to compare the effects of different classifiers on the fault identification results when the feature vectors are the same, a set of training sets and testing sets were arbitrarily selected to classify the features extracted by the HRCMFDE algorithm using ELM, KELM, GA-ELM, and PSO-ELM classifiers, respectively. As can be seen from the figure, the classification effects of the optimized ELM classifiers are all improved, especially the PSO-ELM fault identification algorithm used in this paper which has the highest accuracy. It can guarantee that all of them are correctly classified.

Since the difference in load in the MFPT dataset was not considered, this paper only focused on the diagnosis results of different classification algorithms. The experiments were repeated ten times to take the average identification accuracy and fault identification time for comparison. The results are shown in Table 14. From the table, it can be seen that both GA-ELM and PSO-ELM classifiers can achieve a fault identification accuracy of more than 99%. Unlike the CWRU dataset, the identification accuracy of the PSO-ELM classifier is 99.43%, which is slightly lower than that of the GA-ELM classifier by 0.06%. In a comprehensive analysis, the PSO-ELM classifier still has some advantages in fault identification.

In addition, this paper also compared the identification accuracy of other rolling bearing diagnosis methods using the MFPT dataset for experiments, and the experimental results are shown in Table 15.

Comprehensively analyzing the experimental results of the above two different datasets, the rolling bearing fault diagnosis method based on the HRCMFDE feature extraction algorithm and PSO-ELM classification algorithm proposed in this paper has achieved satisfactory results. Not only the fault identification accuracy but also the fault identification time have good application prospects.

## 5. Conclusions

In this paper, a rolling bearing fault diagnosis method based on HRCMFDE and PSO-ELM is proposed. The HRCMFDE algorithm is proposed for the first time to extract the features of rolling bearing vibration signals. Based on the study of FDE and its variant algorithms, the introduced hierarchical theory algorithm quantifies the high-frequency and low-frequency features of the measured sequence, effectively overcoming the problem of amplitude information loss caused by the coarse-grained process. After extracting the fault features of the vibration signals using the HRCMFDE algorithm, they are inputted to the PSO-optimized ELM classifier for fault identification, which can accurately subdivide different fault types and fault conditions. In this paper, two typical rolling bearing fault datasets are used to verify the effectiveness of the proposed method, and the experimental results show that HRCMFDE has a good descriptive capability for rolling bearing fault features and the proposed fault diagnosis method has satisfactory performance. In addition, this paper further discusses the effectiveness of the proposed method for load migration diagnosis when varying loads are combined to form training sets and testing sets. The results show that the rolling bearing fault diagnosis method proposed in this paper can effectively avoid the algorithm limitations caused by selecting the training set and testing set with the same load, and has a more considerable generalization capability in the application scenarios of the method.

In summary, the rolling bearing fault diagnosis method proposed in this paper has good results. However, the following problems need to be solved in future research. Firstly, it is a challenge to make the algorithm faster and more efficient by reducing its computational complexity to meet the practicality better. Secondly, the extracted fault features can be filtered to improve the level of features and reduce the computational workload by introducing methods such as feature selection. Finally, the paper only focused on the effect of using the entropy algorithm for rolling bearing fault diagnosis. In the future, we will try to select more novel complexity indexes for comparison and research, such as the optimization method based on the LZC algorithm. Meanwhile, we expect that the HRCMFDE feature extraction method proposed in this paper can be applied not only in the field of rolling bearing fault diagnosis, but also to other fields involving nonlinear and non-stationary feature extraction.

## Figures and Tables

**Figure 1 entropy-24-01517-f001:**
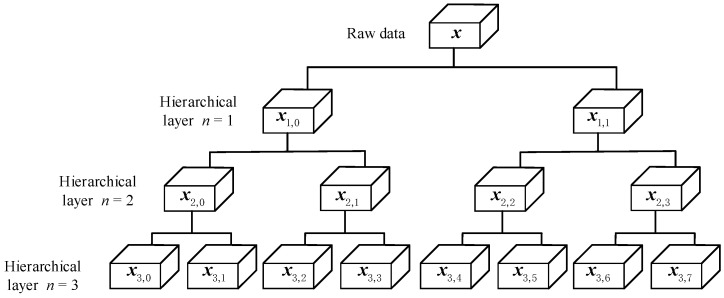
The schematic diagram of hierarchical decomposition at layer *n* = 3.

**Figure 2 entropy-24-01517-f002:**
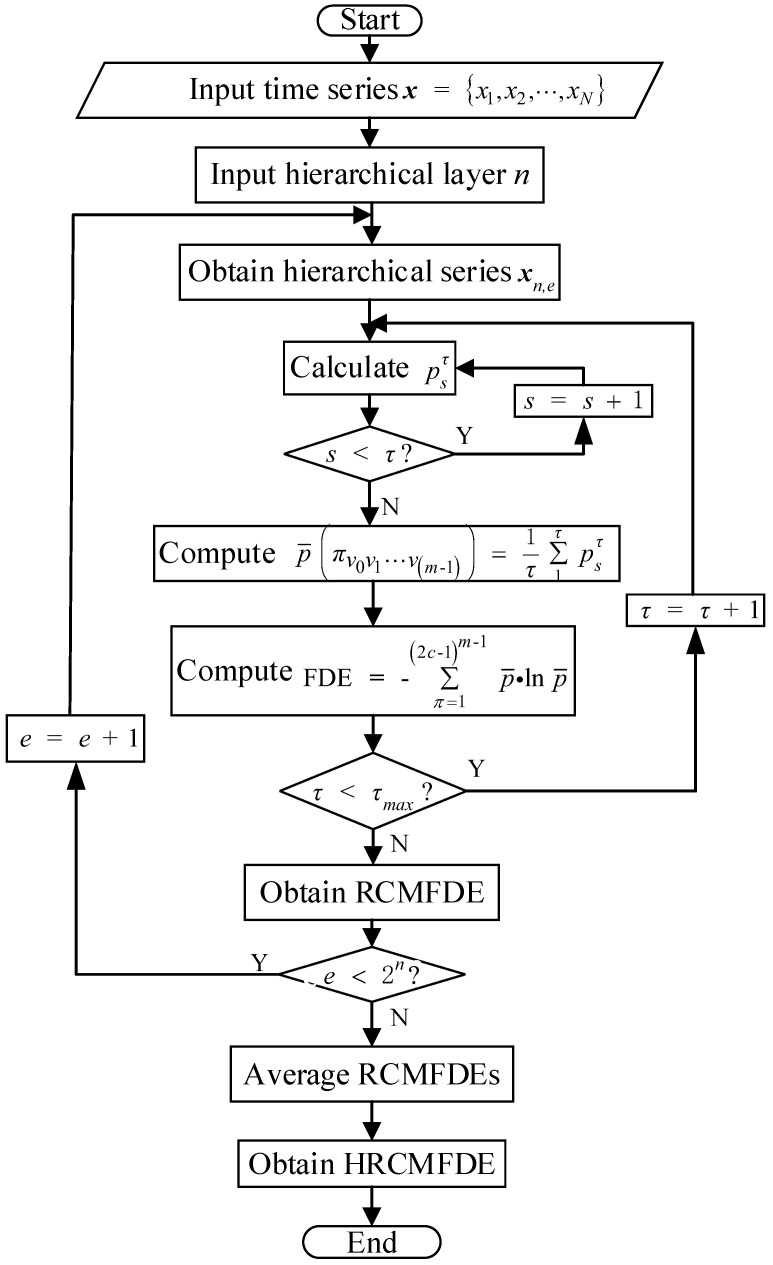
The calculation flowchart of HRCMFDE.

**Figure 3 entropy-24-01517-f003:**
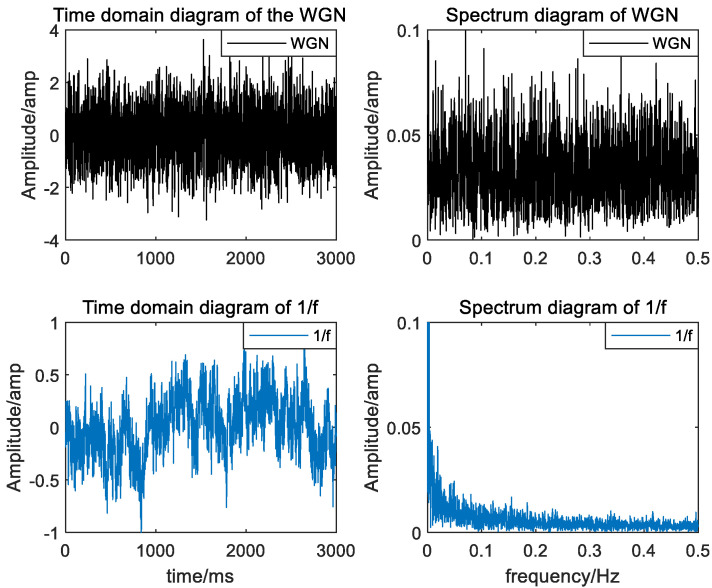
Time domain waveforms and spectrograms of WGN and 1/f noise.

**Figure 4 entropy-24-01517-f004:**
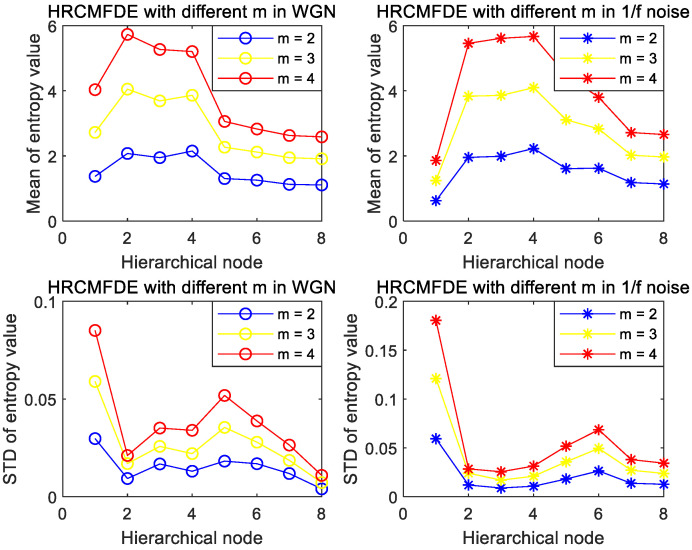
Mean and SD of HRCMFDE for WGN and 1/f noise at different m.

**Figure 5 entropy-24-01517-f005:**
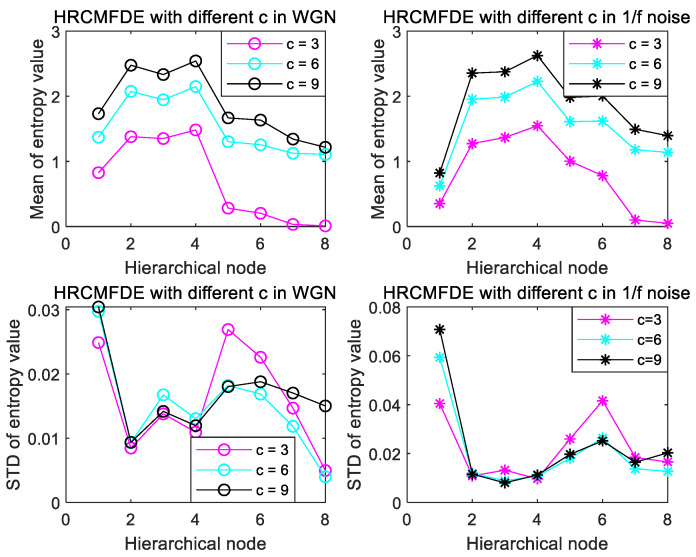
Mean and SD of HRCMFDE for WGN and 1/f noise at different c.

**Figure 6 entropy-24-01517-f006:**
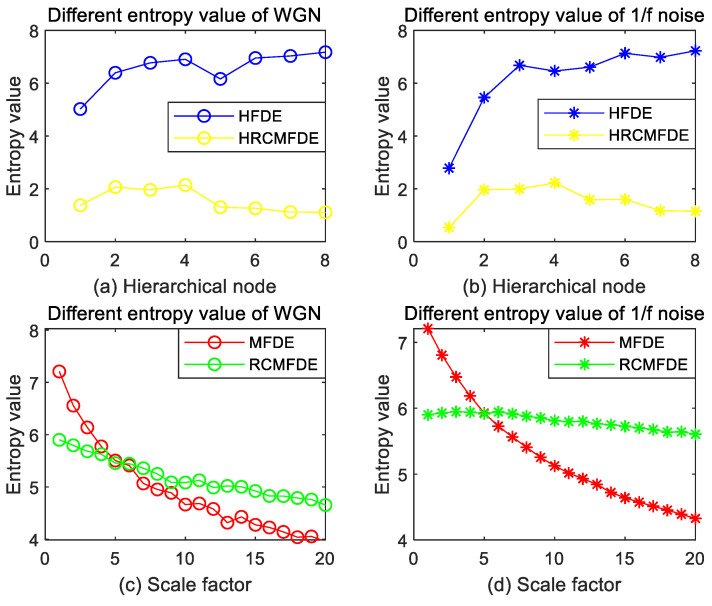
Four different entropy values for WGN and 1/f noise: (**a**,**b**) HFDE and HRCMFDE; (**c**,**d**) MFDE and RCMFDE.

**Figure 7 entropy-24-01517-f007:**
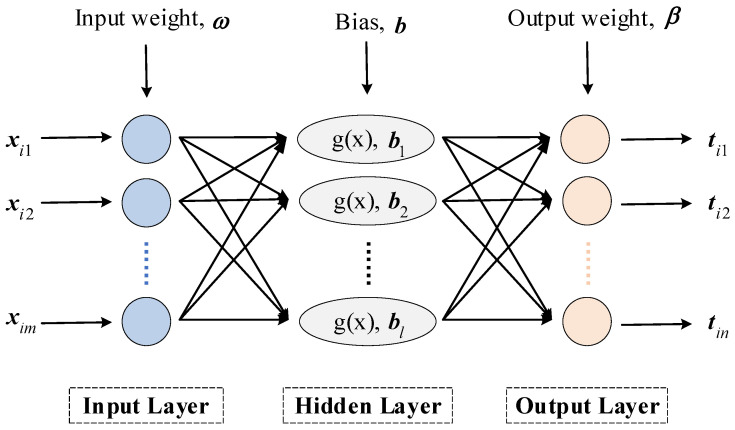
The network structure of ELM.

**Figure 8 entropy-24-01517-f008:**
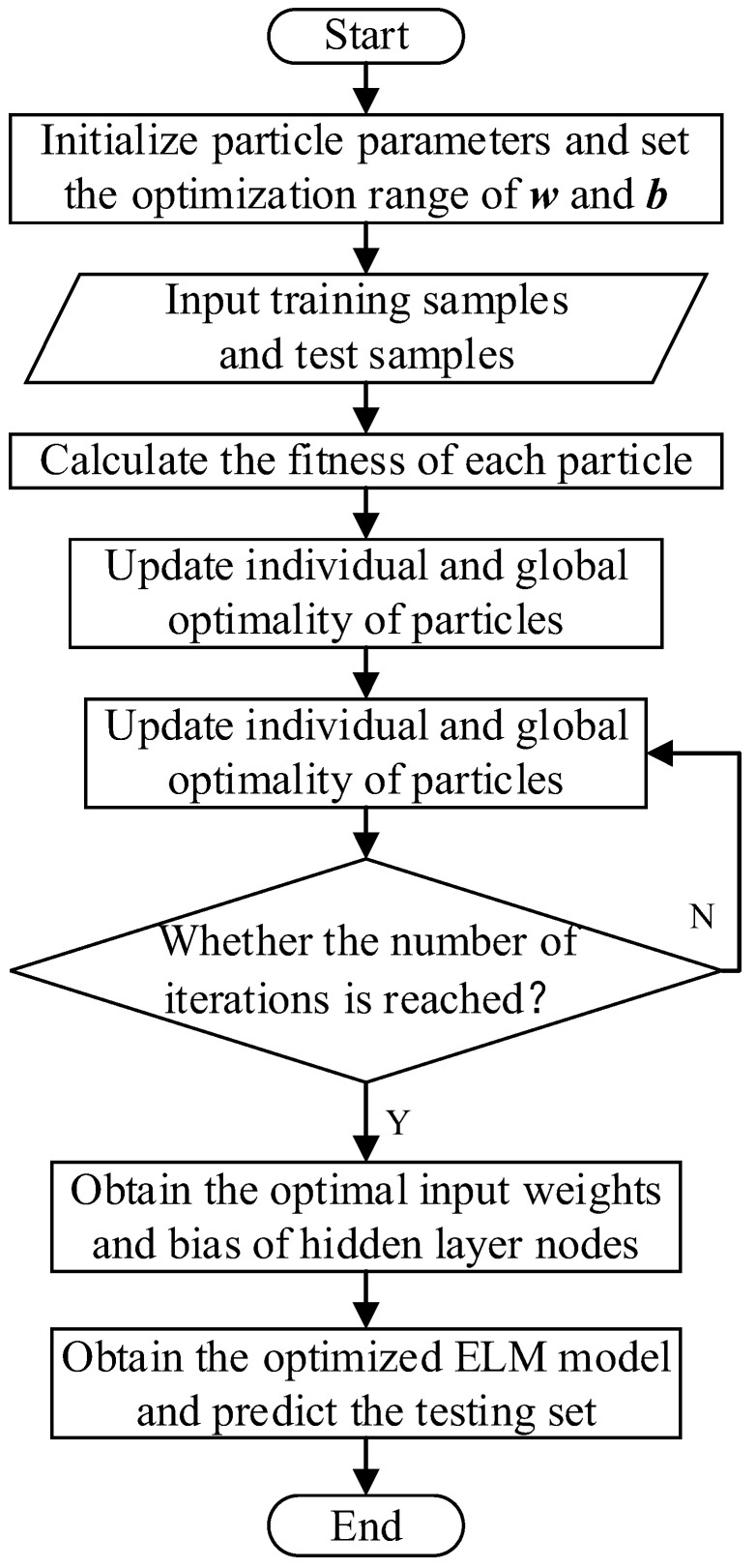
The algorithm flow chart of PSO-ELM.

**Figure 9 entropy-24-01517-f009:**
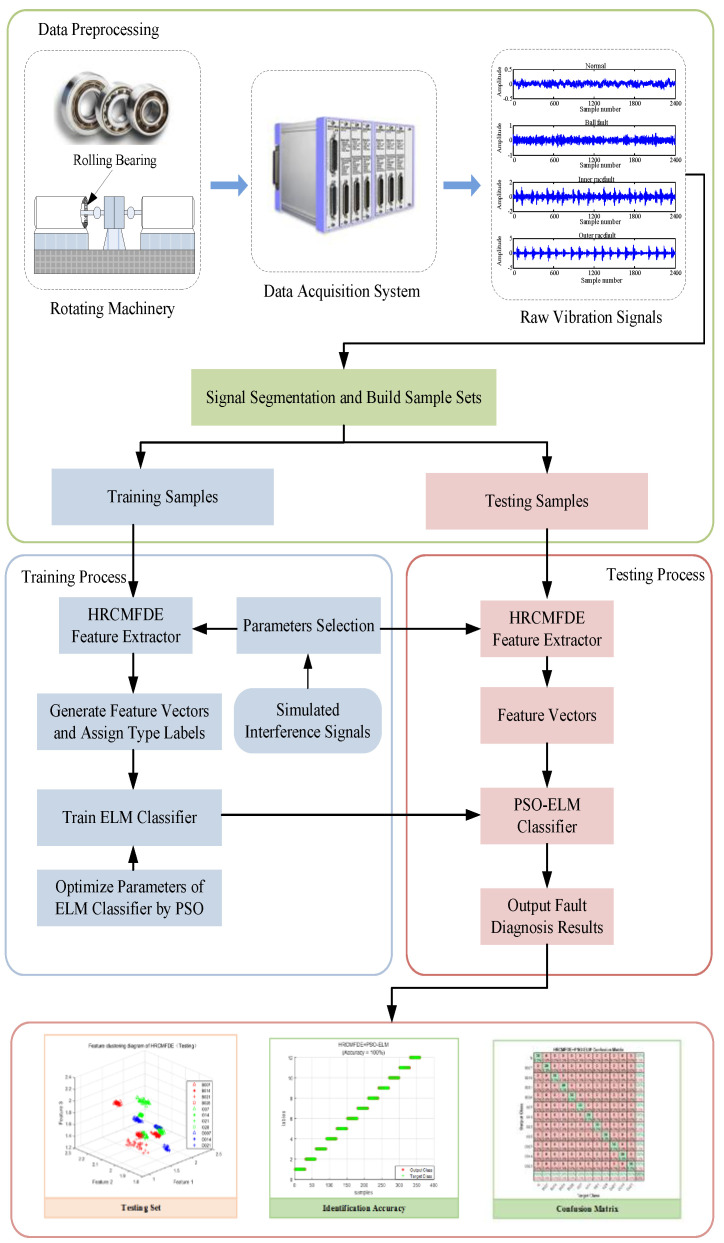
The flowchart of the rolling bearing fault diagnosis method based on HRCMFDE and PSO-ELM.

**Figure 10 entropy-24-01517-f010:**
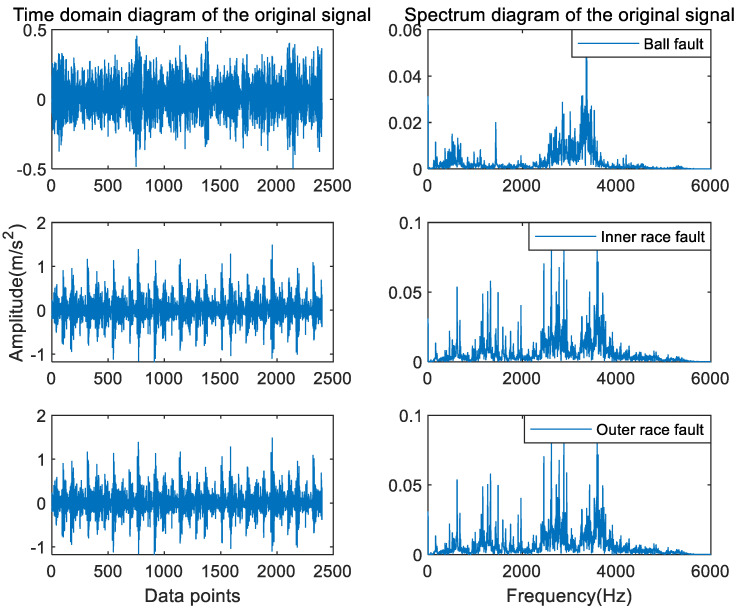
The time domain and spectral diagram of rolling bearing vibration signals with different types in the CWRU dataset.

**Figure 11 entropy-24-01517-f011:**
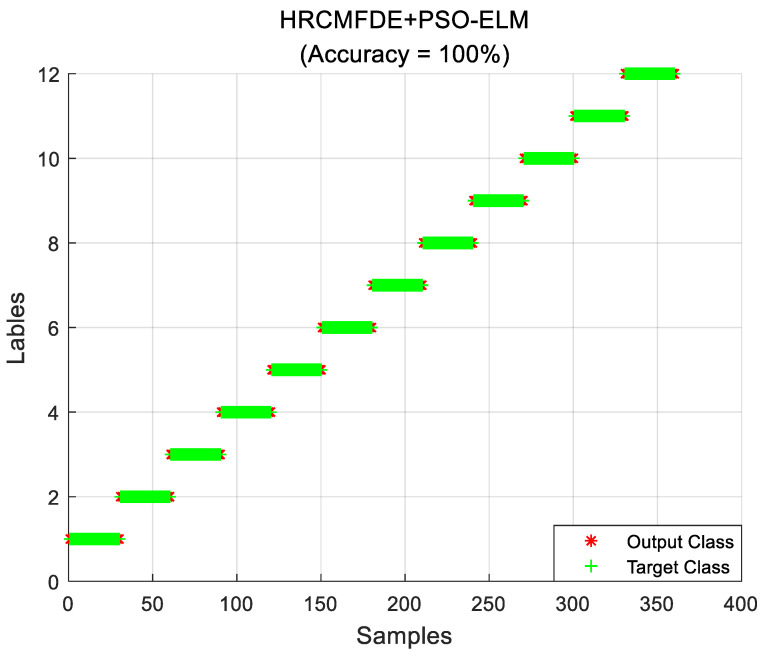
The fault identification results of the proposed method with load 2 hp in the CWRU dataset.

**Figure 12 entropy-24-01517-f012:**
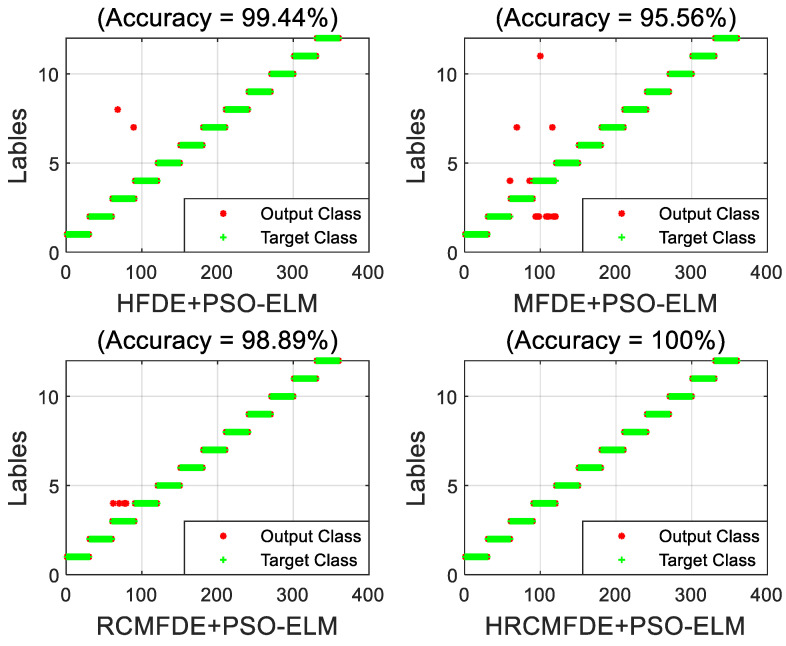
The fault identification results of different feature extraction methods and PSO-ELM with load 0 hp in the CWRU dataset.

**Figure 13 entropy-24-01517-f013:**
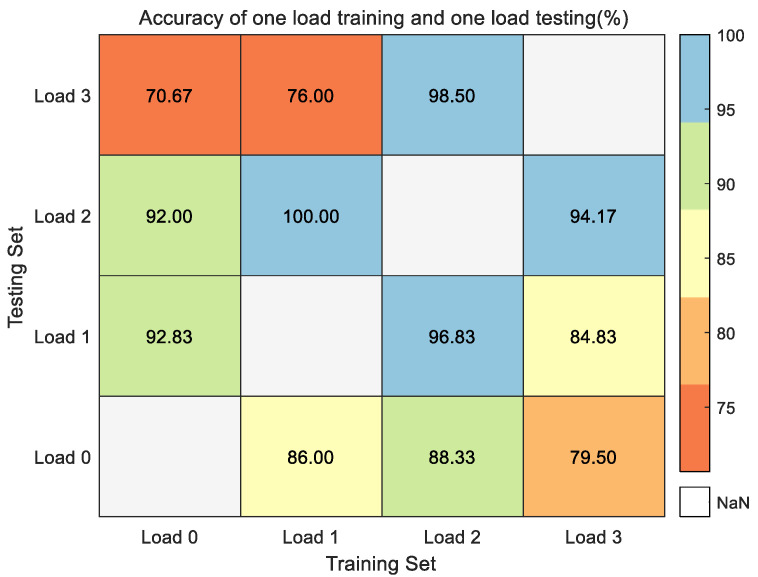
The fault identification accuracy of one load as the training set and another different load as the testing set in the CWRU dataset.

**Figure 14 entropy-24-01517-f014:**
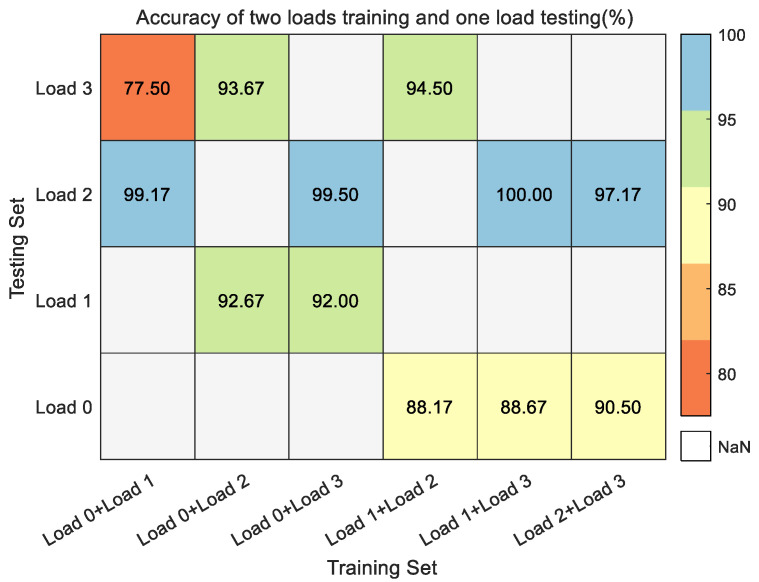
The fault identification accuracy of two loads as the training set and another different load as the testing set in the CWRU dataset.

**Figure 15 entropy-24-01517-f015:**
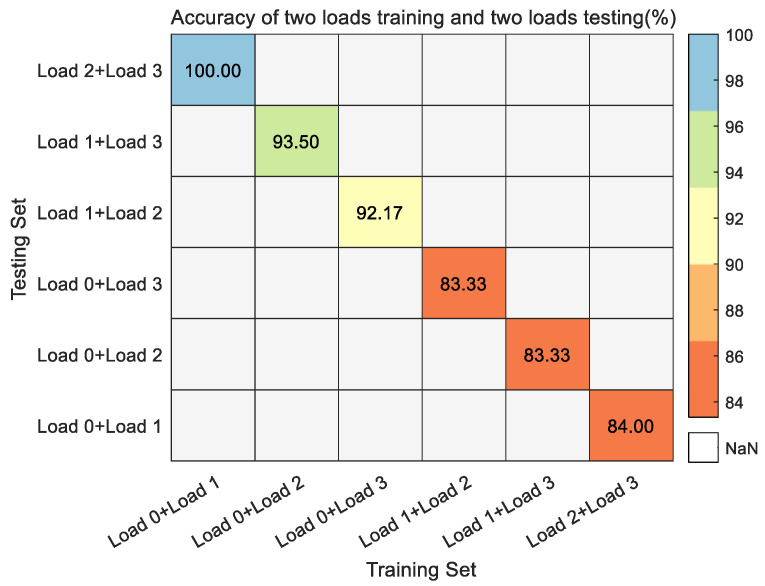
The fault identification accuracy of two loads as the training set and another two varying loads as the testing set in the CWRU dataset.

**Figure 16 entropy-24-01517-f016:**
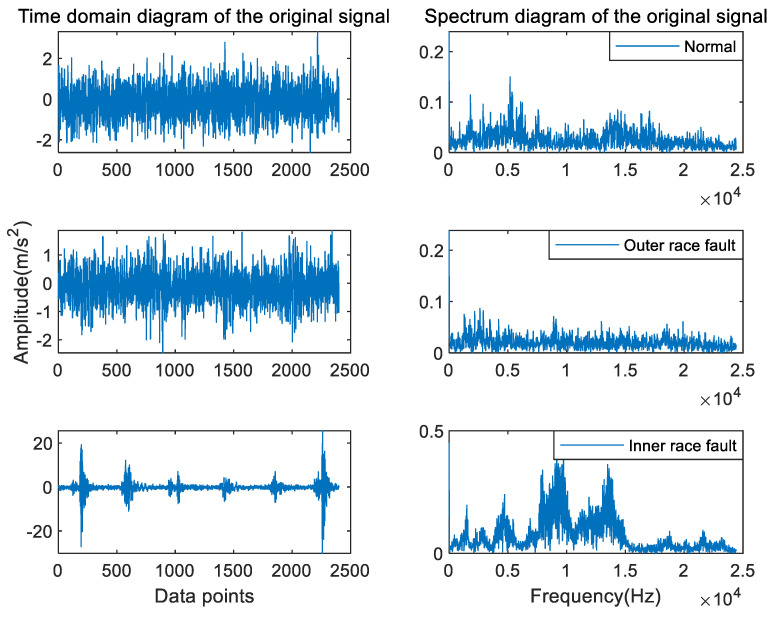
The time domain and spectral diagram of rolling bearing vibration signals with different types in the MFPT dataset.

**Figure 17 entropy-24-01517-f017:**
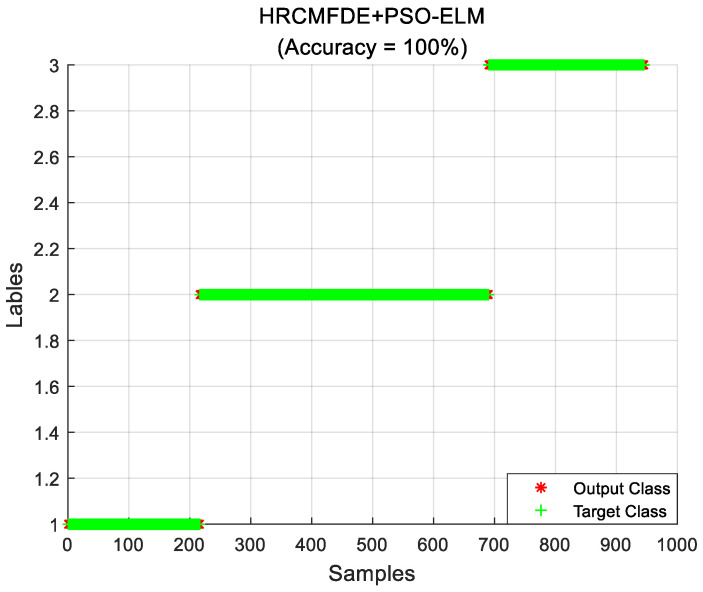
The fault identification results of the proposed method in the MFPT dataset.

**Figure 18 entropy-24-01517-f018:**
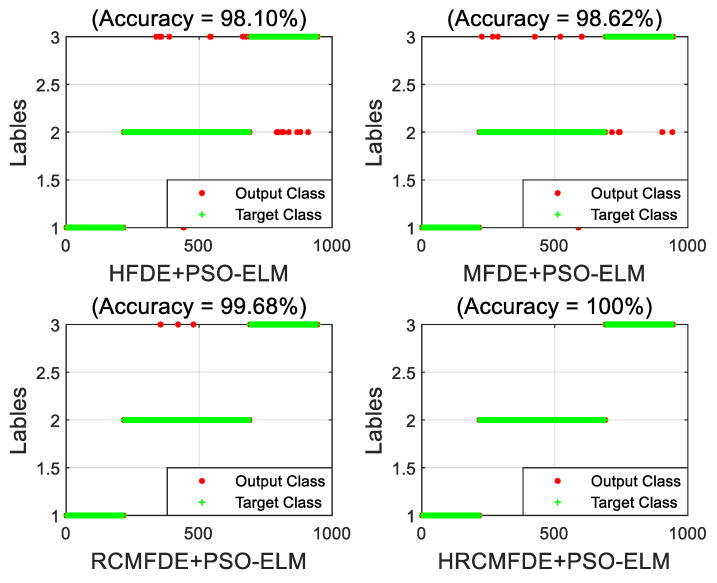
The fault identification results of different feature extraction methods and PSO-ELM in the MFPT dataset.

**Table 1 entropy-24-01517-t001:** CVs of each node of WGN for m at the three cases.

n	1	2	3	4	5	6	7	8
*m* = 2	0.0211	0.0042	0.0066	0.0051	0.0140	0.0134	0.0105	0.0036
*m* = 3	0.0217	0.0045	0.0070	0.0057	0.0156	0.0132	0.0096	0.0038
*m* = 4	0.0211	0.0037	0.0067	0.0065	0.0169	0.0137	0.0101	0.0042

**Table 2 entropy-24-01517-t002:** CVs of each node of 1/f noise for m at the three cases.

n	1	2	3	4	5	6	7	8
*m* = 2	0.0951	0.0061	0.0045	0.0048	0.0112	0.0162	0.0116	0.0112
*m* = 3	0.0972	0.0063	0.0044	0.0051	0.0115	0.0175	0.0134	0.0121
*m* = 4	0.0970	0.0052	0.0045	0.0055	0.0115	0.0180	0.0140	0.0129

**Table 3 entropy-24-01517-t003:** CVs of each node of WGN for c at the three cases.

n	1	2	3	4	5	6	7	8
*c* = 3	0.0301	0.0061	0.0102	0.0074	0.0950	0.1109	0.4438	0.4776
*c* = 6	0.0211	0.0042	0.0066	0.0051	0.0140	0.0134	0.0105	0.0036
*c* = 9	0.0176	0.0038	0.0061	0.0047	0.0108	0.0115	0.0127	0.0123

**Table 4 entropy-24-01517-t004:** CVs of each node of 1/f noise for c at the three cases.

n	1	2	3	4	5	6	7	8
*c* = 3	0.1147	0.0085	0.0096	0.0062	0.0259	0.0533	0.1814	0.3336
*c* = 6	0.0951	0.0061	0.0045	0.0048	0.0112	0.0162	0.0116	0.0112
*c* = 9	0.0492	0.0046	0.0056	0.0037	0.0131	0.0208	0.0123	0.0119

**Table 5 entropy-24-01517-t005:** Parameter selection for HFDE, MFDE, RCMFDE, and HRCMFDE.

Entropy	Embedding Dimension	Number of Classes	Time Delay	Number of Hierarchical Layers	Maximum Scale Factor
HFDE	*m* = 2	*c* = 6	*d* = 1	*n* = 3	\
MFDE	*m* = 2	*c* = 6	*d* = 1	\	*τ _max_* = 20
RCMFDE	*m* = 2	*c* = 6	*d* = 1	\	*τ _max_* = 20
HRCMFDE	*m* = 2	*c* = 6	*d* = 1	*n* = 3	*τ _max_* = 20

**Table 6 entropy-24-01517-t006:** Samples included in the CWRU experimental dataset.

Fault Types	Severity (Inch)	Load (hp)			
		0	1	2	3
Normal	-	√	√	√	√
Ball fault	0.007	√	√	√	√
	0.014	√	√	√	√
	0.021	√	√	√	√
	0.028	√	√	√	√
Inner race fault	0.007	√	√	√	√
	0.014	√	√	√	√
	0.021	√	√	√	√
	0.028	√	√	√	√
Outer race fault	0.007	√	√	√	√
	0.014	√	√	√	√
	0.021	√	√	√	√
	0.028	*	*	*	*

**Table 7 entropy-24-01517-t007:** Sample division and label details of the training set and testing set in the CWRU dataset.

Labels	Fault Types	Abbreviations	Severity (inch)	Number of Training/Testing Samples
1	Normal	N	-	20/30
2	Ball fault	B007	0.007	20/30
3		B014	0.014	20/30
4		B021	0.021	20/30
5		B028	0.028	20/30
6	Inner race fault	I007	0.007	20/30
7		I014	0.014	20/30
8		I021	0.021	20/30
9		I028	0.028	20/30
*10*	Outer race fault	*O007*	0.007	*20/30*
*11*		O014	0.014	20/30
*12*		O021	0.021	20/30

**Table 8 entropy-24-01517-t008:** The accuracy of bearing fault classification with varying loads in the CWRU dataset.

Load	Fault Types	Number of Training Samples	Number of Testing Samples	Average Accuracy
0 hp	N B IR OR	20 80 80 60	30 120 120 90	99.86%
1 hp	N B IR OR	20 80 80 60	30 120 120 90	99.78%
2 hp	N B IR OR	20 80 80 60	30 120 120 90	100%
3 hp	N B IR OR	20 80 80 60	30 120 120 90	100%

**Table 9 entropy-24-01517-t009:** The identification accuracy of different feature extraction algorithms with varying loads in the CWRU dataset.

Feature Extraction	Fault Identification	Load 0 hp	Load 1 hp	Load 2 hp	Load 3 hp	Average Accuracy
HFDE [33]	PSO-ELM	99.72%	100%	100%	99.81%	99.88%
MFDE [28]	PSO-ELM	94.03%	96.69%	99.61%	99.42%	97.44%
RCMFDE [31]	PSO-ELM	98.36%	99.39%	100%	99.92%	99.42%
HRCMFDE	PSO-ELM	99.86%	99.78%	100%	100%	99.91%

**Table 10 entropy-24-01517-t010:** The accuracy and identification time for different fault identification algorithms with varying loads in the CWRU dataset.

Num.	Feature Extraction	Fault Identification	Accuracy (%)	Training Time (s)	Testing Time (s)
1	HRCMFDE	ELM [51]	99.56 ± 0.53	0.0285	0.0085
2	HRCMFDE	KELM [52]	97.94 ± 2.87	0.0046	0.0013
3	HRCMFDE	GA-ELM [53]	99.86 ± 0.16	231.3569	28.6741
4	HRCMFDE	PSO-ELM	99.91 ± 0.11	11.3320	1.2427

**Table 11 entropy-24-01517-t011:** The identification accuracy of other rolling bearing diagnosis methods in the CWRU dataset.

Literature	Feature Extraction	Fault Identification	Number of Classes	Average Accuracy (%)
[54]	WTFD	NPLSSMM	10	99.64
[55]	VMD+MPE	KPCA+CGOA-KELM	4	99.67
[56]	CWT	CNN-SVM	12	98.75
[25]	MAAPE	RF	10	96.00
[57]	EEMD+PE	M-RVM	4	99.58
[11]	RCMFDLZC	DAC	12	96.08
This paper	HRCMFDE	PSO-ELM	12	99.91

**Table 12 entropy-24-01517-t012:** Samples included in the MFPT experimental dataset.

Fault Classes	Fault Types	Load (lb)	Sample Rate (sps)	Sample Time (s)	Data Points
Normal	Baseline	270	97,656	6	585,936
Outer Race Fault	Outer Race Fault	270	97,656	6	585,936
	More Outer Race Fault	25	48,828	3	146,484
		50	48,828	3	146,484
		100	48,828	3	146,484
		150	48,828	3	146,484
		200	48,828	3	146,484
		250	48,828	3	146,484
		300	48,828	3	146,484
Inner Race Fault	Inner Race Fault	0	48,828	3	146,484
		50	48,828	3	146,484
		100	48,828	3	146,484
		150	48,828	3	146,484
		200	48,828	3	146,484
		250	48,828	3	146,484
		300	48,828	3	146,484

**Table 13 entropy-24-01517-t013:** Sample division and label details of the training set and testing set in the MFPT dataset.

Labels	Fault Classes	Number of Total Samples	Number of Training Samples	Number of Testing Samples
1	Normal	360	144	216
2	Outer Race Fault	790	316	474
3	Inner race fault	425	170	255

**Table 14 entropy-24-01517-t014:** The accuracy and identification time for different fault identification algorithms in the MFPT dataset.

Num.	Feature Extraction	Fault Identification	Accuracy (%)	Training Time (s)	Testing Time (s)
1	HRCMFDE	ELM	97.90 ± 0.49	0.0318	0.0198
2	HRCMFDE	KELM	97.71 ± 0.49	0.0371	0.0293
3	HRCMFDE	GA-ELM	99.49 ± 0.48	318.0357	150.4896
4	HRCMFDE	PSO-ELM	99.43 ± 0.38	17.6277	5.1201

**Table 15 entropy-24-01517-t015:** The identification accuracy of other rolling bearing diagnosis methods in the MFPT dataset.

Literature	Feature Extraction	Fault Identification	Number of Classes	Average Accuracy (%)
[56]	CWT	CNN-SVM	3	98.89
[59]	LMD	SNN	3	99.31
[60]	WT	IGoogLeNet	3	99.40
[61]	MSST+SFC-DL	LSVM	3	95.83
[62]	STMSST	CNN	3	98.67
[11]	RCMFDLZC	DAC	3	96.05
This paper	HRCMFDE	PSO-ELM	3	99.43

## Data Availability

The data presented in this study are openly available in reference number [48,58]. The datasets used or analyzed during the current study are available from the corresponding author on reasonable request.

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
