# Peer review of "A Novel Fault Diagnosis Method for Rolling Bearing Based on Hierarchical Refined Composite Multiscale Fluctuation-Based Dispersion Entropy and PSO-ELM"

_entropy, 2022, doi:10.3390/e24111517_

Round 1
Reviewer 1 Report
In this manuscript, the authors applied the techniques of (i) the hierarchical refined composite multiscale fluctuation-based dispersion entropy (HRCMFDE) and (ii) the particle swarm optimization-based extreme learning machine (PSO-ELM) to the problem of rolling bearing fault diagnosis. Experimental results show that the proposed algorithm outperforms the methods. Although the manuscript seems to be well written, I have the following concerns.
(1) In fact, the technique of refined composite multiscale fluctuation-based dispersion has already been adopted in fault diagnosis:
Li, Yuxing, Shangbin Jiao, and Bo Geng. "Refined composite multiscale fluctuation-based dispersion Lempel–Ziv complexity for signal analysis." ISA transactions (2022).
This work should be cited and compared.
(2) The technique of the particle swarm optimization-based extreme learning machine (PSO-ELM) has also been widely applied in fault diagnosis.
Chen, S., Shang, Y., & Wu, M. (2016, June). Application of PSO-ELM in electronic system fault diagnosis. In 2016 IEEE International Conference on Prognostics and Health Management (ICPHM) (pp. 1-5). IEEE.
He, C., Wu, T., Gu, R., Jin, Z., Ma, R., & Qu, H. (2021). Rolling bearing fault diagnosis based on composite multiscale permutation entropy and reverse cognitive fruit fly optimization algorithm–extreme learning machine. Measurement, 173, 108636.
(3) From (1) and (2), I think that the main contribution (also the only contribution) of this work is to combine the two existing techniques for fault diagnosis: HRCMFDE and PSO-ELM. This manuscript has some contribution, but the contribution is not significant.
(4) In Table 15, only three methods were compared. It is insufficient. Moreover, compared to the work in [55], the average accuracy is improved by only 0.03% (from 99.4% to 99,43%). The improvement is not significant.
(5) The manuscript can be much shortened. In Sections 2 and 3, the authors only have to describe the main difference of the proposed method and existing methods. Instead of describing the existing techniques in detail, proper references can be given.
(6) In experiments, figures 10, 12, 13, 15, 17, 21, 23, 24, 26, and 28 and the related text can be removed.
Reviewer 2 Report
In the paper the authors combined two known methods for feature extraction from a time series signal and classification and applied it to the identification of faulty rolling bearings. First a multiscale approach is used to extract features from the signal, then an extreme learning classifier is used to identify and classify the faults. The authors then test their method on two different datasets, CWRU and MFPT, which are public datasets used to benchmark bearing fault classification models. The authors do a good job showing the benefits of their feature extraction method compared to other standard methods both by showing good feature clustering and superior classification results compared to other common feature extraction methods. The model seems to achieve results that are better than state of the art models and is fast to train and make predictions (which is important for online detection) nad robust to noise in the signal (although it would have been nice to see an example of the robustness using noise perturbation to the input signal). While EL has its limitations, it seems that for this classification task and with a limited amount of data it performs extremely well, with the benefit of fast training. Using particle swarm optimization to optimize the input weights and biases gives better results, but the authors should probably explain why not just using more traditional backpropagation gradient based algorithms instead (at this point this is essentially an MLP). If I understand correctly the randomness of the input and hidden layer weight and biases is a vital part of ELM.
Another comment:
Fig 18, 19, 20 are a bit confusing and I am not sure if they are the best way to present the data. The combinations that are not used are assigned 0 accuracy, which is probably not true
Reviewer 3 Report
This paper proposes a novel fault diagnosis method for rolling bearing. The performance of the proposed method is verified by the datasets.
From the results of data validation, this method improves the fault identification accuracy.
Round 2
Reviewer 1 Report
The authors have followed all of my comments to revise their manuscript and performed many new experiments. Although I still think that the method is just a combination the two existing techniques for fault diagnosis: HRCMFDE and PSO-ELM, since this combination has not been done before and the performance is better than the original ones, it is indeed a contribution.
For further revision, the author can summarize the advantages and the limitation of the proposed method, which is helpful for the reader to realize the proposed method.
